# Comparative Evaluation of Growth Performance between Bivalent and Trivalent Vaccines Containing Porcine Circovirus Type 2 (PCV2) and *Mycoplasma hyopneumoniae* in a Herd with Subclinical PCV2d Infection and Enzootic Pneumonia

**DOI:** 10.3390/vaccines9050450

**Published:** 2021-05-03

**Authors:** Hyungmin Um, Siyeon Yang, Taehwan Oh, Keehwan Park, Hyejean Cho, Jeongmin Suh, Kyung-Duk Min, Chanhee Chae

**Affiliations:** 1Department of Veterinary Pathology, College of Veterinary Medicine, Seoul National University, Gwanak-ro 1, Gwanak-gu, Seoul 08826, Korea; uhmhope1@snu.ac.kr (H.U.); didtldus12@gmail.com (S.Y.); Ohth93@gmail.com (T.O.); kylinhp@snu.ac.kr (K.P.); hcho21@snu.ac.kr (H.C.); tobin1210@snu.ac.kr (J.S.); 2Institute of Health and Environment, Seoul National University, 1 Gwanak-ro, Gwanak-gu, Seoul 08826, Korea; fortop@snu.ac.kr

**Keywords:** *Mycoplasma hyopneumoniae*, porcine circovirus type 2, trivalent vaccine

## Abstract

The present field trial compared two combined vaccines of porcine circovirus type 2 (PCV2) and *Mycoplasma hyopneumoniae*, each administered in herd with subclinical PCV2d infection and enzootic pneumonia. One vaccine was a bivalent containing PCV2a and *M. hyopneumoniae* and the other was a trivalent vaccine containing PCV2a and 2b (PCV2a/b), and *M. hyopneumoniae*. The defining difference between these two vaccines was the inclusion or absence of PCV2b antigen. A total of 480, 21day-old pigs were randomly allocated to one of four treatment groups (120 pigs per group, male = 60 and female = 60). These groups included; one-dose trivalent-vaccinated, two-dose trivalent-vaccinated, one-dose bivalent-vaccinated, and unvaccinated. The one- and two-dose trivalent vaccinated pigs exhibited significantly better growth performance when compared with those vaccinated with the bivalent vaccine. The one- and two-dose trivalent vaccinated pigs also reduced the amount of PCV2d loads in the blood and feces, and resulted in a lower *M. hyopneumoniae* load in the larynx when compared with one-dose bivalent vaccinated pigs. Statistical differences were not observed between the one- and two-dose trivalent-vaccinated groups in terms of growth performance, serology, amount of PCV2d loads in the blood and feces, amount of *M. hyopneumoniae* load in larynx, and pathological lesions. The results of the present study will provide swine practitioners and producer with comparative clinical field data to select the proper vaccine and vaccination regiment for herds suffering from subclinical PCV2d infection and enzootic pneumonia.

## 1. Introduction

Porcine circovirus type 2 (PCV2), a member of the family Circoviridae, is a common virus of pigs found throughout the world and is recognized as one of the most economically threatening pathogens to the global pork industry [1]. PCV2 may not be a new virus but it still remains a constant challenge due to the wide range of syndromes and diseases that is causes. Postweaning multisystemic wasting syndrome (PMWS), porcine dermatitis and nephropathy syndrome (PDNS), porcine respiratory disease complex (PRDC), reproductive failure, and enteric manifestations are all examples of porcine circovirus-associated diseases (PCVAD). Although PCV2 has been classified as a well-controlled pathogen since 2018, due to the wide use of vaccines, most farms still experience subclinical PCV2 infection [2]. Currently, PCV2 is classified into at least eight genotypes that are designated consecutively based on the time of first identification with lower case letters, “a to h” [3]. The “d” genotype (PCV2d) is considered the most prevalent and predominant genotype in Asia and North America, today [4,5,6].

*Mycoplasma hyopneumoniae* is prevalent and highly contagious in the majority of swine herds throughout the global pig industry. *M. hyopneumoniae* causes mycoplasmal pneumonia, which is characterized by a chronic, non-productive cough with high morbidity and low mortality. Enzootic pneumonia that is caused by *M. hyopneumoniae* when combined with opportunistic bacteria, such as *Pasteurella multocida*, continues to be a significant chronic respiratory disease [7]. Enzootic pneumonia leads to decreased average daily gain and an increased number of days to market weight, both of which result in significant economic losses [7].

PCV2 and *M. hyopneumoniae* are economically important pathogens and the primary agents involved in the PRDC found within global pig production systems. Vaccination for PCV2 and *M. hyopneumoniae* is one of the most effective strategies in the control of both pathogens, especially Asian pork industry [8]. Korean swine farms currently use combination vaccines containing PCV2 and *M. hyopneumoniae* for more than 50% of their pigs (http://www.kahpha.or.kr (accessed on 29 April 2021)). Combined vaccination is consequently considered part of routine management practices.

Recently, a new trivalent vaccine containing PCV2a and 2b (PCV2a/b), and *M. hyopneumoniae* (Fostera^®^ Gold PCV MH/CircoMax^®^ Myco, Zoetis, Parsippany, NJ, USA) was introduced into the global market (http://www.zoetisus.com (accessed on 29 April 2021)). The PCV2b antigen of the trivalent vaccine is of particular interest as it is genetically closely related to PCV2d (formerly referred to as mutant PCV2b), which is currently the predominant PCV2 genotype in Asian pig populations [4,5,6]. Although PCV2a-based vaccines can provide cross-protection against PCV2d under experimental conditions [9,10,11,12], the emergence of PCV2d has still been linked to PCVAD outbreaks within these PCV2a-vaccinated herds [13,14,15]. In an additional comparative experimental study, PCV2b-based vaccines may be less effective than PCV2a-based vaccines at protecting against the PCV2d genotype [16]. Nevertheless, comparative field trial between a bivalent vaccine containing PCV2a and *M. hyopneumoniae* and a trivalent vaccine containing PCV2a/b and *M. hyopneumoniae* has yet to be undertaken. The objective of this study was to compare a bivalent and trivalent vaccine, with an emphasis on the evaluation of growth performance in herds in the presence of subclinical PCV2d infection and enzootic pneumonia.

## 2. Materials and Methods

### 2.1. Farm History

The clinical field trial was conducted on a 1200-sow, farrow-to-finish swine farm that implemented an all-in-all-out production system. The farm was selected based on its history of subclinical PCV2d infection and enzootic pneumonia. The status of porcine reproductive and respiratory syndrome virus (PRRSV) at the farm was stable; with no active PRRSV circulation (high-parity sows were the only seropositive animals in the herd). No PRRS modified-live virus vaccine was administered for at least one year in sows and piglets. Piglets were vaccinated for PCV2 (Ingelvac CircoFLEX^®^, Boehringer Ingelheim Vetmedica Inc., St. Joseph, MO, USA) and *M. hyopneumoniae* (Ingelvac MycoFLEX^®^, Boehringer Ingelheim Vetmedica Inc.) at 3 weeks of age. Submitted cases met the definition of subclinical PCV2 infection [17] based on decreased average daily gain without overt clinical signs, no or minimal histopathological lesions in inguinal lymph nodes, and the presence of low amounts of PCV2 in inguinal lymph nodes by immunohistochemistry in 3 out of 5 suspected pigs. In addition, *M. hyopneumoniae* infection was determined in all three, 68 day-old pigs by display of severe dry coughing, histopathological peribronchiolar lymphoid tissue hyperplasia, and the detection of *M. hyopneumoniae* in lung samples by real-time PCR [18]. A pilot survey was implemented to assess the circulation of PCV2 and *M. hyopneumoniae* in the herd. Pre-trial investigations identified a PCV2 serological profile that presented an increase in antibody titers starting around 7 weeks of age. Pigs that were 7–15 weeks of age tested positive for PCV2 in their blood by PCR methodology. *M. hyopneumoniae* serology tested as partially positive in 7 week-old pigs, and completely positive in 10 week-old pigs. Together, these results show early and prolonged PCV2 and *M. hyopneumoniae* infections were circulating within the herd.

### 2.2. Experimental Design

To minimize sow variation, eight, 21 day-old pigs were randomly selected using the random number generator function (Excel, Microsoft Corporation, Redmond, WA, USA) from each sow and assigned evenly (two pigs per sow) to each of the four groups. A total of 480 pigs was randomly divided into 4 groups (120 pigs per group; male =60 and female = 60) using the same software and function (Table 1). The pigs in the VacA1 group were intramuscularly vaccinated with a 2.0 mL dose of the trivalent vaccine (Fostera^®^ Gold PCV MH, Serial No: 413369A, Expiration date: 03 February 2022, Zoetis, Parsippany, NJ, USA) at 21 days of age. The pigs in the VacA2 group were intramuscularly vaccinated with a 1.0 mL dose of the trivalent vaccine (Fostera^®^ Gold PCV MH) at 21 and 42 days of age, respectively. Pigs in the VacB group were intramuscularly vaccinated with a 2.0 mL dose of the bivalent vaccine (Porcilis^®^ PCV M Hyo, Lot No. C746B02, Expiration date: 09 September 2021, MSD Animal Health, Boxmeer, Netherlands) at 21 days of age. Pigs in the UnVac group were injected intramuscularly with 2.0 mL of phosphate buffered saline (PBS, 0.01 M, pH 7.4) at 21 days of age. Pigs were comingled and randomly assigned into 48 pens within the same building. Each pen contained 10 pigs with a similar proportion of each treatment per pen. Pens were identical in design and equipment which included free access to a feed and water trough.

Whole blood, and fecal and laryngeal swabs were collected at 0 (21 days old), 28 (49 days old), 49 (70 days old), 91 (112 days old) days post-vaccination (dpv). Pigs were snared and restrained with a mouth gag for laryngeal swab collection. Swabs were guided with a laryngoscope down into the larynx. The internal walls of the laryngeal cartilages were then swept with the swabs once the larynx was visualized and the epiglottis was in a low position [19].

### 2.3. Clinical Observations

The pigs were monitored daily for abnormal clinical signs and scored weekly using scores ranging from 0 (normal) to 6 (death) [20]. Observers were blinded to vaccination and type of vaccine status. Mortality rate was calculated as the number of pigs that died divided by the number of pigs initially assigned to that group within batch. Pigs that died or were culled throughout the study was necropsied. Evaluation of injection site reaction including palpation was performed 24 h post-vaccination.

### 2.4. Average Daily Weight Gain

The live weight of each pig was measured at 0 (21 days old), 49 (70 days old), and 154 (175 days old) days post-vaccination. The average daily weight gain (ADWG; gram/pig/day) was analyzed over two time periods: (i) between 21 and 70 days old and (ii) between 70 and 175 days old. ADWG during the different production stages was calculated as the difference between the starting and final weight divided by the duration of the stage. Data for dead pigs were included in the calculation.

### 2.5. T Cell Epitope Contents Comparison Analysis

PCV2d strain (SNUVR202002, GenBank no. MW821481) was isolated in inguinal lymph node from 68 day-old pig in submitted diagnostic case. The relatedness between vaccine sequences and field strain was analyzed by T cell epitope contents comparison (EpiCC) analysis as previously described [21]. To quantify vaccine T cell epitope coverage, the shared EpiCC score of each vaccine-field strain comparison was divided by that field strain’s baseline EpiCC and expressed as a percentage.

### 2.6. Quantification of PCV2d DNA in Blood and Feces

DNA was extracted from serum and fecal samples using the commercial kit (QIAamp DNA Mini Kit, QIAGEN, Valencia, CA, USA) to quantify PCV2d genomic DNA copy numbers by real-time PCR [22].

### 2.7. Quantification of M. hyopneumoniae DNA in Laryngeal Swabs

DNA was extracted from laryngeal swabs using the commercial kit (QIAamp DNA Mini Kit, QIAGEN) to quantify the *M. hyopneumoniae* genomic DNA copy numbers by real-time PCR [18].

### 2.8. Serology

The serum samples were tested using the commercially available enzyme-linked immunosorbent assay (ELISA) kits for *M. hyopneumoniae* (M. hyo. Ab test, IDEXX Laboratories Inc. Inc., Westbrook, ME, USA) and PCV2 (SERELISA PCV2 Ab Mono Blocking, Synbiotics, Lyon, France). Serum samples were considered positive for *M. hyopneumoniae* antibody if the sample-to-positive (S/P) ratio was ≥0.4, and positive for anti-PCV2 antibodies if the reciprocal ELISA titer was >350, in accordance with the manufacturer’s instructions for each kit.

### 2.9. Pathology

The severity of macroscopic lung lesions was scored to estimate the percentage of the lung affected by pneumonia. The scoring was done by two pathologists (Chae and one graduate student) at the Seoul National University (Seoul, Republic of Korea). For the entire lung (100 points were assigned as follows; 10 points each to the right cranial lobe, right middle lobe, left cranial lobe, and left middle lobe, 27.5 points each to the right caudal lobe and left caudal lobe, and 5 points to the accessory lobe) [20]. Two blinded veterinary pathologists then examined the collected lung and lymphoid tissue sections and scored the severity of peribronchiolar lymphoid tissue hyperplasia by mycoplasmal pneumonia lesions (0 to 6) [23]. Lymphoid lesion severity was scored (0 to 5) based on lymphoid depletion and granulomatous inflammation [24].

### 2.10. Statistical Analysis

Prior to statistical analysis, real-time PCR data were transformed to log_10_ values. Statistical analyses were performed IBM SPSS Statistics for Windows version 23.0 (IBM Corp., Armonk, NY, USA). The Shapiro–Wilk test will be utilized to test the collected data for a normal distribution. One-way analysis of variance (ANOVA) was used to examine whether there are statistically significant differences at each time point within different groups. A one-way ANOVA test result with such a statistical significance was be further evaluated by conduction a post-hoc test for a pairwise comparison with Tukey’s adjustment. If the normality assumption was not met, the Kruskal–Wallis test was be performed. Results from Kruskal–Wallis test which showed statistical significance were further evaluated with the Mann–Whitney test to include Tukey’s adjustment to compare the differences among the groups. Results were reported in *p*-value where a value of *p* < 0.05 was considered to be significant.

## 3. Results

### 3.1. Clinical Signs

Respiratory signs, such as dyspnea and tachypnea, were significantly lower (*p* < 0.05) in vaccinated animals (VacA1, VacA2, and VacB groups) than those in unvaccinated animals (UnVac group) at 21 to 126 dpv. A comparison between vaccinated groups determined that respiratory signs, such as dyspnea and tachypnea in the VacA1 group were significantly lower (*p* < 0.05) than those in the VacB group at 63 and 98 dpv.

### 3.2. Average Daily Weight Gain

A difference in mean body weight was not observed between vaccinated (VacA1, VacA2, and VacB groups) and unvaccinated (UnVac group) animals at the time the study began (21 days of age). The ADWG of vaccinated animals (VacA1, VacA2, and VacB groups) was significantly higher (*p* < 0.05) than that of unvaccinated animals (UnVac group) during the fattening period (70 to 175 days of age) and overall period (21 to 175 days). In a comparison of vaccinated groups, the ADWG of the VacA1 group was significantly higher (*p* < 0.05) than that of the VacB group during the fattening period (70 to 175 days of age). The ADWG of the VacA1 and VacA2 groups was significantly higher (*p* < 0.05) than that of the VacB group during the overall period (21 to 175 days of age) (Table 2).

### 3.3. Mortality

Diagnostic results indicated that mortality was primarily related to co-infection with PCV2 and *M. hyopneumoniae* in unvaccinated animals. In the VacA1 group, one pig died of unknown hemorrhagic diarrhea at 63 days of age. Two additional pigs from the VacA1 group died of bronchopneumonia as determined by a combination of *M. hyopneumoniae* as detected with PCR, and *P. multocida* that was isolated from the lungs at 72 and 75 days of age. In the VacA2 group, three pigs died of bronchopneumonia, as determined by a combination of PCV2d that was detected with PCR, and *Glaesseralla parasuis* that was isolated from the lungs at 52, 70, and 78 days of age. Three pigs in the VacB group died of bronchopneumonia, as determined by a combination of *M. hyopneumoniae* that was detected with PCR, and *Trueperella pyogenes* that was isolated from the lungs at 60, 80, and 82 days of age. Two additional VacB pigs died of bronchopneumonia, as determined by a combination of PCV2d that was detected with PCR and *P. multocida* that was isolated from the lungs at 72 days of age. In the UnVac group, one pigs died of salmonellosis, as determined by *Salmonella typhimurium* that was isolated from the large intestine at 52 days of age. Four UnVac group pigs died of bronchopneumonia, as determined by a combination of PCV2d and *M. hyopneumoniae* that were detected with PCR, and *G. parasuis* that was isolated from the lungs at 64, 75, 88, and 110 days of age. Three additional UnVac group pigs died of bronchopneumonia from a combination of *M. hyopneumoniae* that was detected with PCR, and *P. multocida* and *T. pyogenes* that were isolated from the lungs at 72 (2 pigs) and 80 days of age.

### 3.4. T Cell Epitope Content Comparison Analysis

Shared EpiCC score was higher in the trivalent vaccine compared to the bivalent vaccine. T cell epitope coverage of bivalent vaccine against field PCV2 strain (SNUVR202002) was 62% and of trivalent vaccine against the same field PCV2d strain was 83%. This represented 33% improvement of an epitope coverage (Table 3).

### 3.5. Quantification of PCV2d DNA in Blood and Feces

The amount of PCV2d DNA loads in blood from vaccinated animals (VacA1, VacA2, and VacB groups) were significantly lower (*p* < 0.05) than that of unvaccinated animals (UnVac group) at 28, 49, and 91 dpv. PCV2d DNA loads in blood from the VacA1 and VacA2 groups were significantly lower (*p* < 0.05) than that of the VacB group at 49 dpv (Figure 1A). The amount of PCV2d DNA loads in feces from vaccinated animals (VacA1, VacA2, and VacB groups) were significantly lower (*p* < 0.05) than that of unvaccinated animals (UnVac group) at 28, 49, and 91 dpv (Figure 1B).

### 3.6. Quantification of M. hyopneumoniae DNA in Laryngeal Swabs

The amount of *M. hyopneumoniae* DNA loads in laryngeal swabs from vaccinated animals (VacA1, VacA2, and VacB groups) were significantly lower (*p* < 0.05) than that of unvaccinated animals (UnVac group) at 28, 49, and 91 dpv. In comparison of vaccinated groups, the amount of *M. hyopneumoniae* DNA loads in laryngeal swabs from the VacA1 group was significantly lower (*p* < 0.05) than that of the VacB group at 49 dpv (Figure 2).

### 3.7. Immune Responses against PCV2

Vaccinated animals from VacA1, VacA2, and VacB groups produced significantly higher (*p* < 0.05) PCV2 ELISA titers at 28, 49, and 91 dpv than that of unvaccinated animals from the UnVac group. In comparison of vaccinated groups, the PCV2 ELISA titers of the VacA1 and VacA2 groups was significantly higher (*p* < 0.05) than that of the VacB group at 49 and 91 dpv (Figrue 3A).

### 3.8. Immune Responses against M. hyopneumoniae

Vaccinated animal from VacA1, VacA2, and VacB groups produced significantly higher (*p* < 0.05) *M. hyopneumoniae* ELISA S/P ratios at 28, 49, and 91 dpv than that of unvaccinated animal from the UnVac group. In comparison of vaccinated groups, the *M. hyopneumoniae* ELISA S/P ratios of the VacA1 and VacA2 groups was significantly higher (*p* < 0.05) than that of the VacB group at 91 dpv (Figure 3B).

### 3.9. Pathology

Vaccinated (VacA1, VacA2, and VacB) groups had significantly lower (*p* < 0.05) macroscopic and microscopic lung lesion scores when compared to unvaccinated (UnVac) group at 154 dpv (Table 2).

## 4. Discussion

The present field trial is used to compare two different combination vaccines; a bivalent vaccine containing PCV2a and *M. hyopneumoniae*, and a trivalent vaccine containing PCV2a/b and *M. hyopneumoniae*. Under the field conditions of the present study, where subclinical PCV2d infection and enzootic pneumonia was circulating within the farm, pigs vaccinated with the trivalent vaccine exhibited significantly better growth performance when compared with the bivalent vaccine. There were no statistical differences in growth performance between one-dose and two-dose trivalent-vaccinated groups. The economic benefit of trivalent-vaccinated groups over bivalent-vaccinated group was evaluated by differences on market weight at the time of slaughter. Trivalent-vaccinated groups improved significantly (*p* < 0.05) body weight by 1.245 kg/pig (106.495 kg in combined trivalent vaccinated group vs. 105.25 kg from bivalent-unvaccinated group), leading to an increase of revenue by 2.98 US dollars (exchange rate; US $1.00 = 1169.40 Korean Won) per pig.

The improved growth performance of the trivalent-vaccinated groups over the bivalent-vaccinated group may be attributed to the different epitope determinant between vaccine and field PCV2 strain; T cell epitope coverage of bivalent vaccine against field PCV2d strain was 62% and of trivalent vaccine against the same field PCV2d strain is 83%. Therefore, trivalent PCV2a/b and *M. hyopneumoniae* vaccine strains provide better protection against field PCV2d strain compared to bivalent vaccine containing PCV2a vaccine strain only. These results agree with previous EpiCC analysis, in which combination of PCV2a and PCV2b vaccine shared on average more T cell epitope content with strains from all the different genotypes than monovalent PCV2a vaccines [21]. PCV2a-based vaccines provide partial cross-protection against PCV2d under experimental conditions [9,10,11,12]. Several differences exist between these experimental challenge conditions of the study from those of commercial pig farms. Pigs in commercial farms are continuously exposed and re-exposed to the prevalent field PCV2d virus by horizontal transmission. Natural confections as well as other intrinsic and extrinsic factors also exacerbate disease in less-controlled commercial setting. Under field conditions, the levels of cross-protection provided by PCV2a-based vaccines against PCV2d have been questioned, due to reports of PCV2d identification in PCV2a-vaccinated herds [13,14,15]. In this comparative field trial, trivalent vaccination reduced the amounts of PCV2d loads in the blood and feces when compared to bivalent vaccination. The reduction of PCV2 viremia is well correlated with protection against PCV2 infection [25,26,27]. The present results indicate that the trivalent vaccination provided better protection against PCV2d when compared to bivalent vaccination against PCV2d subclinical infection under field conditions.

The strains of the *M. hyopneumoniae* antigen and adjuvant formulation differed between the two combination vaccines. In particular, adjuvant formulation is known to affect the immunogenicity and protective effect of inactivated whole-cell *M. hyopneumoniae* bacterins [28]. Trivalent vaccination reduced the amount of *M. hyopneumoniae* load in larynx when compared to the bivalent vaccine. Although correlation between the reduction of *M. hyopneumoniae* in the larynx and vaccine protection is not well known, reducing the amount of *M. hyopneumoniae* loads in the larynx are more likely to reduce horizontal transmission to neighboring pigs.

Regardless of vaccine type, vaccinated animals had a significantly greater reduction in mycoplasmal lung lesions compared to unvaccinated animals. These results are consistent with previous studies, where vaccination of pigs with *M. hyopneumoniae* reduces pneumonic lung lesions in field trials [29,30,31]. A significant difference in lymphoid lesions was not observed between the vaccinated and unvaccinated groups. This may be attributed to the subclinical PCV2 infection on the farm where the field clinical trial was conducted, as PCV2-associated lymphoid lesions are typically mild in pigs with subclinical PCV2 infection [17].

This is the first comparative field trial that evaluated the differences between a bivalent PCV2a and *M. hyopneumoniae* vaccine and trivalent PCV2a/b and *M. hyopneumoniae* vaccine. Broader coverage resulting from the PCV2 vaccine’s two genotypes provides additional insurance against the evolving PCV2 virus in the field. It is clinically meaningful to conduct comparative field clinical trial on farm with subclinical PCV2d infection and enzootic pneumonia.

## 5. Conclusions

This is the first comparative field trial that evaluated the differences between a bivalent and trivalent vaccine containing PCV2a and *M. hyopneumoniae* and a trivalent vaccine containing PCV2a/b and *M. hyopneumoniae*. Pigs vaccinated with the trivalent vaccine exhibited significantly better growth performance when compared with the bivalent vaccine. No statistical differences in growth performance were observed between one-dose and two-dose trivalent-vaccinated groups. The improved growth performance of the trivalent-vaccinated groups over the bivalent-vaccinated group may be attributed to the different epitope determinant between vaccine and field PCV2 strain; T cell epitope coverage of bivalent vaccine against field PCV2d strain is 62% and of trivalent vaccine against the same field PCV2d strain is 83%. It is clinically meaningful to conduct comparative field clinical trial on farm with PCV2d subclinical infection and enzootic pneumonia.

## Figures and Tables

**Figure 1 vaccines-09-00450-f001:**
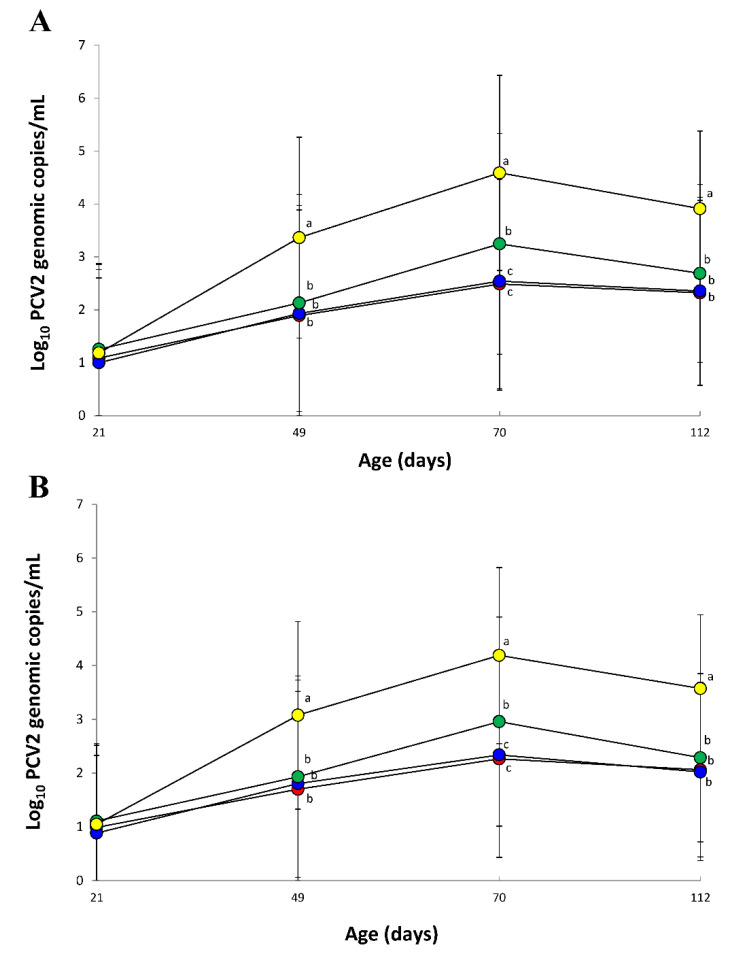
Mean values of the genomic copy number of PCV2d DNA in serum (**A**) and feces (**B**) from VacA1 (●), VacA2 (●), VacB (●), and UnVac (●). Variation is expressed as the standard deviation. Different superscripts (a, b, and c) indicate significant (*p* < 0.05) different among 4 groups.

**Figure 2 vaccines-09-00450-f002:**
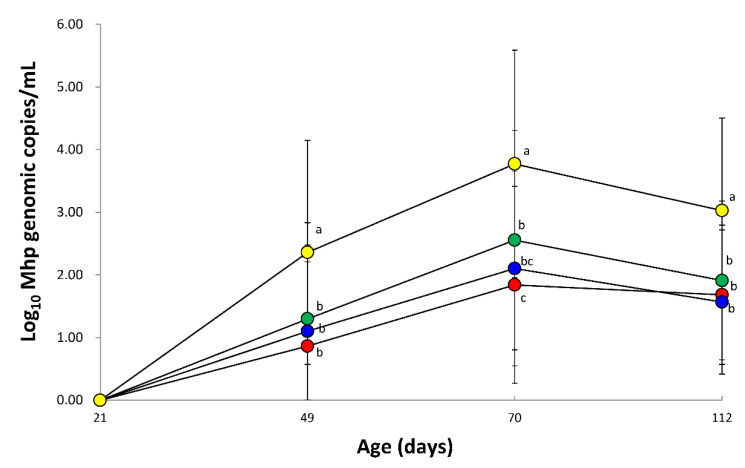
Mean values of the genomic copy number of *Mycoplasma hyopneumoniae* DNA in laryngeal swab from VacA1 (●), VacA2 (●), VacB (●), and UnVac (●). Variation is expressed as the standard deviation. Different superscripts (a, b, and c) indicate significant (*p* < 0.05) different among 4 groups.

**Figure 3 vaccines-09-00450-f003:**
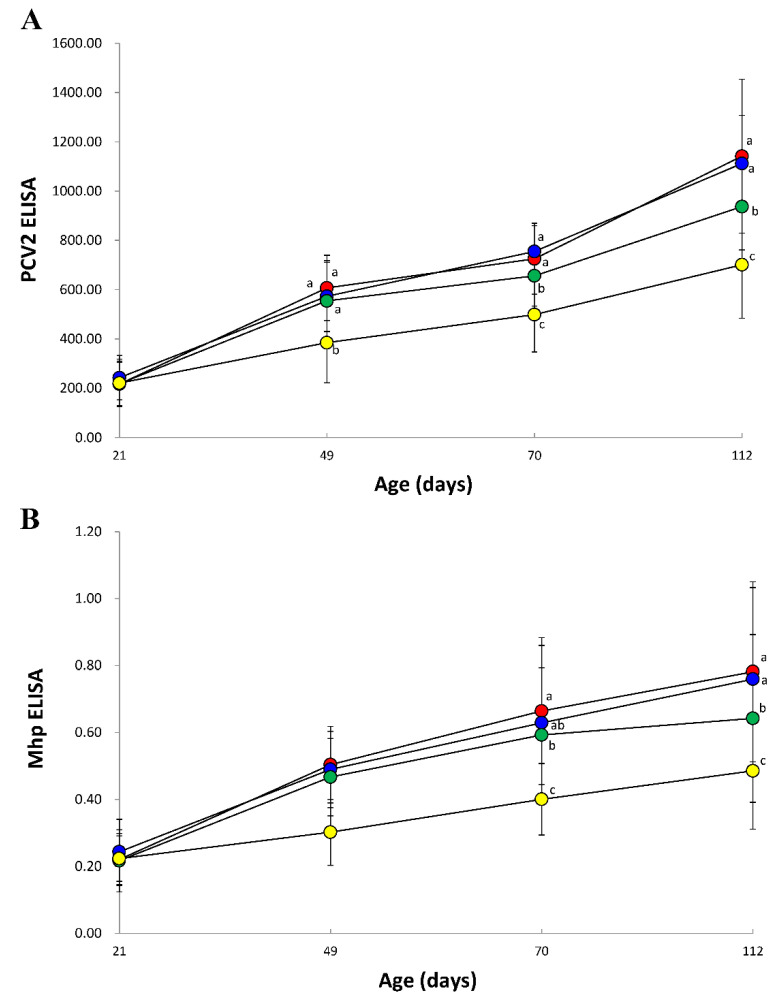
Mean values of the anti-PCV2 antibodies (**A**) and anti-*Mycoplasma hyopneumoniae* antibodies (**B**) from VacA1 (●), VacA2 (●), VacB (●), and UnVac (●). Variation is expressed as the standard deviation. Different superscripts (a, b, and c) indicate significant (*p* < 0.05) different among 4 groups.

**Table 1 vaccines-09-00450-t001:** Field experimental design.

Groups	No. of Pigs	Vaccine	Dosage	Age (Days)
VacA1	120	Fostera^®^ Gold PCV MH	One (2.0 mL)	21
VacA2	120	Fostera^®^ Gold PCV MH	Two (1.0 mL)	21, 42
VacB	120	Porcilis^®^ PCV M Hyo	One (2.0 mL)	21
UnVac	120	Phosphate buffered saline	One (2.0 mL)	21

**Table 2 vaccines-09-00450-t002:** Growth performance with average daily weight gain (ADWG) and pathology between vaccinated and unvaccinated animals.

	Age	Groups			
	(Days)	VacA1	VacA2	VacB	UnVac
ADWG	21–70	399.90 ± 25.44	401.89 ± 24.05	395.51 ± 24.20	393.57 ± 31.13
(gram/pig/day)	70–175	775.62 ± 20.65 ^a^	772.73 ± 18.45 ^a,b^	765.23 ± 22.73 ^b^	715.74 ± 26.26 ^c^
	21–175	656.06 ± 11.85 ^a^	654.74 ± 11.38 ^a^	647.65 ± 14.17 ^b^	613.46 ± 14.33 ^c^
Body weight	21	5.57 ± 0.32	5.56 ± 0.33	5.51 ± 0.35	5.50 ± 0.36
	175	106.60 ± 1.82 ^a^	106.39 ± 1.71 ^a^	105.25 ± 2.15 ^b^	99.96 ± 2.19 ^c^
Macroscopic	175	17.82 ± 6.90 ^a^	18.21 ± 7.85 ^a^	19.70 ± 8.21 ^a^	28.60 ± 10.67 ^b^
lung lesions					
Microscopic	175	0.73 ± 0.56 ^a^	0.78 ± 0.60 ^a^	0.88 ± 0.65 ^a^	2.04 ± 0.93 ^b^
lung lesions					
Microscopic	175	0.69 ± 0.59	0.73 ± 0.61	0.86 ± 0.58	1.07 ± 0.37
lymphoid lesions					

^a,b,c^ Different superscripts indicate significant (*p* < 0.05) difference among 4 groups.

**Table 3 vaccines-09-00450-t003:** Summary of T cell epitope contents comparison (EpiCC) scores between porcine circovirus type 2 (PCV2) vaccine and field strain.

	ORF2 of PCV2 of Vaccines		
	Monovalent ^a^	Bivalent ^b^	Trivalent ^c^
Vaccine baseline ^d^	6.83	6.50	8.66
Average baseline (sd) ^e^	10.49 (0.16)		
EpiCC ^f^	6.83	6.50	8.66
Coverage ^g^	65.36%	62.22%	82.82%

^a^ Monovalent (Ingelvac CircoFLEX^®^) vaccine used in the farms. ^b^ Bivalent (Porcilis^®^ PCV M Hyo) vaccine used in this study. ^c^ Trivalent (Fostera^®^ Gold PCV MH) vaccine used in this study. ^d^ EpiCC score calculated for the vaccine compared to itself. ^e^ Average baseline EpiCC score (and standard deviation) of full-length field strain. ^f^ EpiCC score of the vaccine compared to full-length field strain. ^g^ Coverage of each field strain’s baseline EpiCC score expressed as a percentage.

## Data Availability

The data present in the study are available on request from the corresponding author.

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
