# Peer review of "Comparative Evaluation of Growth Performance between Bivalent and Trivalent Vaccines Containing Porcine Circovirus Type 2 (PCV2) and Mycoplasma hyopneumoniae in a Herd with Subclinical PCV2d Infection and Enzootic Pneumonia"

_vaccines, 2021, doi:10.3390/vaccines9050450_

Round 1

Reviewer 1 Report

Line 28 - delete "will" in the statement. 

Line 36 - put reference after "pathogens to the global pork industry"

Line  57- put reference for "the most effective strategies in the control of both pathogens, especially Asian pork industry."

Line 63 - add reference for "was introduced into the global market. "

Line  94- put reference after "and the detection of M. hyopneumoniae in lung samples by real-time PCR. "

Add animal ethic number if you have it or explain this study do not need animal ethic because of .....

Line  120- did you collect blood swab samples or whole blood or both? 

Line  189- Clarify "Respiratory signs" in the statement.

Line  233-  is it 33% improvement?

Line  345- reword "No statistical differences in growth performance were observed between one-dose and two-dose trivalent-vaccinated groups." 

Author Response

Response to Reviewer #1’s comments

Line 36 - put reference after "pathogens to the global pork industry"

Response: Author added the reference

Line  57- put reference for "the most effective strategies in the control of both pathogens, especially Asian pork industry."

Response: Author added the reference

Line 63 - add reference for "was introduced into the global market. "

Response: Author added the wbsite for the information of vaccine product.

Line  94- put reference after "and the detection of M. hyopneumoniae in lung samples by real-time PCR. "

Response: Author added the reference,

Add animal ethic number if you have it or explain this study do not need animal ethic because of .....

Response: Author added animal ethic number in Institutional Review Board Statement section.

Line  120- did you collect blood swab samples or whole blood or both? 

Response: Authors clarified the collection of sample.

Line  189- Clarify "Respiratory signs" in the statement.

Response: Author clarified "Respiratory signs" in the statement.

Line  233-  is it 33% improvement?

Response: Authors clarified 33% improvement.

Line  345- reword "No statistical differences in growth performance were observed between one-dose and two-dose trivalent-vaccinated groups." 

Response: Authors rephrased “No statistical differences in growth performance were observed between one-dose and two-dose trivalent-vaccinated groups." 

Reviewer 2 Report

A comparison between different vaccination regimes was performed in a herd affected by PCV2d and M. hyopneumoniae. Two vaccine formulations were used one with PCV2a and M. hyopneumoniae and one with PCV2a/b and M. hyopneumoniae. The latter vaccine was either given once (day 21) or twice (days 21 and 42) whereas the PCV2b/M. hyopnumoniae vaccine only was given once (day21).  The health status of the pigs was recorded together with average daily weight gain and the serological response to M. hyopneumoniae and PCV2, respectively was determined by ELISA. The presence of microbial DNA was determined by qPCR in relevant tissue samples and pathological lesions were scored at autopsy. In addition a T cell epitope content coparison analysis was performed. The results were compared between groups and compared to those in pigs in the control group, only injected with PBS.

In all aspects the vaccinated pigs performed better than the unvaccinated. The pigs vaccinated twice showed a better growth performance and less lung lesions than those only vaccinated once. The serological response in unvaccinated pigs confirmed the presence of both PCV2 and M. hyopneumoniae at a herd level albeit this slow and steady increase in serum antibodies was at a lower magnitude that in those vaccinated. In general, pigs vaccinated with PCV2a/b and M. hyopneumoniae developed higher antibody responses than those given PCV2b and M. hyopneumoniae. Surprisingly, no clear effect of the booster vaccination is seen in the antibody response to PCV2. It is suggested that the better performance of the pigs vaccinated with PCV2/b is attributed to cross-protection against PCV2d via T-cell immunity. This can however not explain the significantly lower response to M. hyopneumoniae in pigs receiving “Vac B”. The authors thus need to discuss that the difference in effect between the two vaccines can be attributed to other components in the vaccine such as the adjuvant. In general the manuscript is well written and data clearly presented.

Line 167: “were be”, remove “be”

Author Response

Response to Reviewer #2’s comments

A comparison between different vaccination regimes was performed in a herd affected by PCV2d and M. hyopneumoniae. Two vaccine formulations were used one with PCV2a and M. hyopneumoniae and one with PCV2a/b and M. hyopneumoniae. The latter vaccine was either given once (day 21) or twice (days 21 and 42) whereas the PCV2b/M. hyopnumoniae vaccine only was given once (day21).  The health status of the pigs was recorded together with average daily weight gain and the serological response to M. hyopneumoniae and PCV2, respectively was determined by ELISA. The presence of microbial DNA was determined by qPCR in relevant tissue samples and pathological lesions were scored at autopsy. In addition a T cell epitope content coparison analysis was performed. The results were compared between groups and compared to those in pigs in the control group, only injected with PBS.

In all aspects the vaccinated pigs performed better than the unvaccinated. The pigs vaccinated twice showed a better growth performance and less lung lesions than those only vaccinated once. The serological response in unvaccinated pigs confirmed the presence of both PCV2 and M. hyopneumoniae at a herd level albeit this slow and steady increase in serum antibodies was at a lower magnitude that in those vaccinated. In general, pigs vaccinated with PCV2a/b and M. hyopneumoniae developed higher antibody responses than those given PCV2b and M. hyopneumoniae. Surprisingly, no clear effect of the booster vaccination is seen in the antibody response to PCV2. It is suggested that the better performance of the pigs vaccinated with PCV2/b is attributed to cross-protection against PCV2d via T-cell immunity.

This can however not explain the significantly lower response to M. hyopneumoniae in pigs receiving “Vac B”. The authors thus need to discuss that the difference in effect between the two vaccines can be attributed to other components in the vaccine such as the adjuvant.

Response: Authors added the discussion that the difference in effect between the two vaccines van be attributed to other components in the vaccine such as the adjuvant.

Line 167: “were be”, remove “be”

Response: Authors removed “be” in Line 167.

Reviewer 3 Report

This is a well written and presented manuscript detailing a comparison of PCV2 (a and d) and M hyo combined vaccines in a field study.

The methods are well detailed and described, and the study design is robust.

The results are clearly presented, with useful figures and tables (although it would be useful to describe the significance of the superscript letters in table 2).

The discussion is comprehensive and clearly frames the significance of the work in the field, including comparisons with other published work, and with supporting references as required.

The conclusions are supported by the data presented in the manuscript.

Author Response

Response to Reviewer #3’s comments

The results are clearly presented, with useful figures and tables (although it would be useful to describe the significance of the superscript letters in table 2).

Response: Authors added the description about the significance of the superscript letter in table 2.

The discussion is comprehensive and clearly frames the significance of the work in the field, including comparisons with other published work, and with supporting references as required.

Response: This is the first comparison study for combination vaccine of PCV2 and Mycoplasma hyopneumoniae. Authors added one reference to supporting the comparison of Mycoplasmal vaccine due to adjuvant formulation.
